# Barrier-free reverse-intersystem crossing in organic molecules by strong light-matter coupling

Yi Yu[1,2], Suman Mallick[1,2], Mao Wang [1] & Karl Börjesson [1✉]

Strong light-matter coupling provides the means to challenge the traditional rules of chemistry. In particular, an energy inversion of singlet and triplet excited states would be fundamentally remarkable since it would violate the classical Hund's rule. An organic chromophore possessing a lower singlet excited state can effectively harvest the dark triplet states, thus enabling 100% internal quantum efficiency in electrically pumped light-emitting diodes and lasers. Here we demonstrate unambiguously an inversion of singlet and triplet excited states of a prototype molecule by strong coupling to an optical cavity. The inversion not only implies that the polaritonic state lies at a lower energy, but also a direct energy pathway between the triplet and polaritonic states is opened. The intrinsic photophysics of reversed-intersystem crossing are thereby completely overturned from an endothermic process to an exothermic one. By doing so, we show that it is possible to break the limit of Hund's rule and manipulate the energy flow in molecular systems by strong light-matter coupling. Our results will directly promote the development of organic light-emitting diodes based on reversed-intersystem crossing. Moreover, we anticipate that it provides the pathway to the creation of electrically pumped polaritonic lasers in organic systems.

[1] Department of Chemistry and Molecular Biology, University of Gothenburg, Gothenburg, Sweden. [2] These authors contributed equally: Yi Yu, Suman Mallick
✉email: karl.borjesson@gu.se

In organic materials, the triplet state ($T_1$) always possesses a lower energy than the corresponding singlet state ($S_1$) due to subtle electron-electron interactions. This phenomena is commonly referred as Hund's rule[1], and it results in energy transfer from $T_1$ to $S_1$ being an endothermic process. It is often desired to manipulate the rate of energy transfer between states. For example, in electrically pumped light emitting systems, singlet and triplet excited states are formed in a 1:3 ratio based on spin statistics[2]. The triplet state acts as an energy loss channel as the electronic transition to the ground state is forbidden due to spin conservation restrictions. This leads to a maximum 25% internal quantum efficiency of the system, limiting the overall efficiency.

In recent years, thermally activated delayed fluorescence (TADF) has aroused tremendous interest as it harvests the energy of triplet states through reversed-intersystem crossing (RISC) to the corresponding emissive singlet state. This process requires a small energy gap ($\Delta E$) between $T_1$ and $S_1$. Thanks to tremendous efforts in synthetic chemistry, organic molecules with very small $\Delta E$ (<0.2 eV) are now routinely made[3,4]. Nonetheless, the decrease of $\Delta E$ is usually accompanied with a reduction of spin–orbit or vibronic coupling, which has a positive correlation with the rate of RISC[5–7]. Thus, the overall RISC dynamics are not often significantly improved in this way.

Strong light-matter coupling has emerged as a new tool for tailoring molecular properties without touching the chemical structure[8,9]. The formed hybrid states, referred to as polaritons, play key roles in modifying the photophysical and photochemical processes in the strong coupling regime[10–13], such as the enhancement of Förster type and vibrational energy transfer[14–17], tilting the ground-state reactivity landscape[18,19], facilitating Bose–Einstein condensation and organic lasing[20–25], reducing energy losses in photovoltaics[26–29], manipulating triplet state dynamics[30–33], maximizing superconducting current[34] and optimization of artificial photosynthesis[35]. In particular, we have previously demonstrated polariton-enhanced RISC in optical cavities by reducing the energy gap between singlet (polariton) and triplet states[36], the method of which was then adopted to achieve an energy inversion of the two states[37]. However, it was found that the RISC dynamic showed negligible differences compared to the bare molecular case.

Herein, these seemingly contradictory results are rationalized, and a system with a barrier-free reversed-intersystem crossing is achieved. We begin by coupling a TADF molecule strongly to the vacuum electromagnetic field by placing it inside an optical cavity. We then show that the formed polaritonic state is energetically inverted with the molecular triplet state by measuring prompt and delayed emission. Next, given the Arrhenius type of analysis of the temperature dependence of RISC rates, a direct and barrier free conversion between triplet and polaritonic states is found. Finally, the coupling between the molecular centered triplet states and the hybrid polaritonic state seems to depend on the constitution of the polariton. The connection is lost with a too low matter contribution to the lower polariton, which could be an explanation to earlier observations.

## Results and discussion
### Photophysics of thermal activated delayed fluorescence.
The conversion of triplet states to the corresponding singlet state is a thermally activated process. TADF molecules are therefore generally engineered to have a small intrinsic $S_1$–$T_1$ energy difference ($\Delta E$)[38]. The possibility of a $S_1$–$T_1$ energy level inversion in the strong coupling regime therefore depends on the interplay between the coupling strength and $\Delta E$ (Fig. 1a). In an optical cavity, light couples to the Frank–Condon state of an organic molecule at a magnitude proportional to the transition dipole

moment associated to the state being coupled and the square root of the molecular concentration. In this study, pristine molecular films were used as to maximize the molecular concentration, and thus the light matter coupling strength. Furthermore, a small Stokes shift is desired, as it indicates little energy relaxation from the Frank-Condon state to the relaxed first excited singlet state. Five TADF molecules were evaluated in light of their potential to invert their triplet-singlet energy levels: 3DPyM-pDTC[39], DABNA-1, DABNA-2[40], TBN-TPA[41], and a boron difluoride curcuminoid derivative[42] (Supplementary Fig. 1 for structures and Supplementary Note 1 for synthesis). DABNA-2 was deemed to be the best compromise on $\Delta E$ vs. coupling strength, attributed to its large absorption coefficient, small Stokes shift and well-defined absorption envelope (see "Methods" section, Supplementary Figs. 2–4 and Supplementary Table 1). Figure 1b shows the absorption and prompt and delayed emission from a neat DABNA-2 film. Gated emission at 77 K showed weak residual fluorescence as well as phosphorescence at longer wavelengths (~515 nm). The long-lived fluorescence is a result of RISC from the excited triplet to singlet states by thermal activation. These two processes could therefore be distinguished from each other by observing their temperature dependencies (vide infra). The $\Delta E$ value derived from these measurements was 123 meV, which was consistent with previous studies of 1 wt% films (140 meV)[40]. We will later show that the small delayed fluorescence shifted to lower energies, and was therefore greatly enhanced when the molecule was strongly coupled to an optical cavity.

### Energy inversion in the strong coupling regime.
Having settled the energy level alignment of pristine DABNA-2 films, we turned our attention to the ability of strong light-matter coupling to perturb this alignment. Strong coupling occurs between a molecular excited state and a cavity mode, when the exchange of energy between these is larger than energy dissipation. It is manifested by a splitting of the molecular excited state into two polaritonic states, $P^+$ and $P^-$ (Fig. 1a), which are separated in energy by the Rabi splitting ($\hbar\Omega_R$). To reach the strong coupling regime, neat DABNA-2 films were placed in between two Ag mirrors forming a Fabry-Pérot cavity (Supplementary Fig. 5). Figure 2a–d display angle-resolved reflectivity data of four such cavities with different thickness of the molecular film. In all the cases, we observed a clear anti-crossing of the two newly emerged polaritonic branches. Based on a coupled harmonic oscillators approach[43], we extracted Rabi splittings of 420, 420, 440, and 400 meV for Cavity 1 to Cavity 4, respectively. From the full width at half maximum (FWHM) of the molecular absorbance (189 meV) and cavity resonance (<150 meV), we saw that the strong coupling regime was reached in all cavities. Moreover, by the varying thickness of the active layer in these cavities, the energy of $P^-$ gradually shifted across $T_1$ and a complete energy inversion was achieved in Cavity 3 ($E_{P^-} - E_{T1} = -68$ meV) and Cavity 4 ($E_{P^-} - E_{T1} = -148$ meV). However, an energy inversion does not guarantee a spontaneous energy flow from $T_1$ to $P^-$. As polaritons are hybrid light-matter states[44], the relative excitonic contribution to $P^-$ decreased from Cavity 1 to Cavity 4 (Supplementary Fig. 6). We will later show that the polariton composition is of key importance for achieving a spontaneous and barrier-free energy transfer between the molecular centered $T_1$ states and the delocalized $P^-$ state.

The energy inversion in the strong coupling regime was further verified by probing the emission from the polaritonic and triplet states (Fig. 2e–h). Phosphorescence from Cavity 1, Cavity 2, and Cavity 4 was located at the same position as that of the neat film, which clearly showed that the triplet state was not perturbed when the singlet excited state was strongly coupled to the cavity.

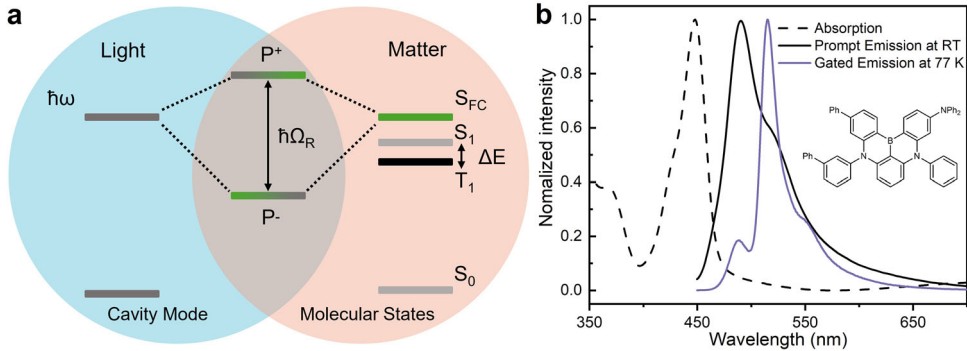

**Fig. 1 Strong coupling system for realizing $S_1$–$T_1$ energy inversion. a** Jablonski diagram presenting the energy gap ($\Delta E$) between first excited singlet ($S_1$) and triplet ($T_1$) states. Strong coupling of the Frank–Condon state ($S_{FC}$) with a resonant cavity mode ($\hbar\omega$) leads to the formation of two hybrid light-matter states, $P^+$ and $P^-$, which are separated in energy by the Rabi splitting ($\hbar\Omega_R$). **b** Absorption (dashed black), prompt emission (solid black, at rt) and gated emission (purple, 100 ns gate delay, at 77 K, delayed fluorescence at ~490 nm and phosphorescence at 515 nm) spectra of a neat film.

It should be noted that the presence of non-resonant emission was made possible by the top mirror of the cavity being semi-transparent (Supplementary Fig. 7). For Cavities 1 and 2, the $P^-$ emission was higher in energy compared to phosphorescence, thus confirming that the $P^-$ state was higher in energy than the $T_1$ state. As for Cavity 3, in which a complete inversion was achieved between the polaritonic and triplet state, only emission from the polaritonic state was observed. The disappearance of phosphorescence indicated a potential energy flow to the lower energy polaritonic state. However, for Cavity 4, in which the polaritonic state lies well below the triplet state, the majority of the emission was phosphorescence, which suggested that there was no energy flow directly to the polaritonic state. In the next section, we will discuss the temperature-dependent dynamics of the intersystem crossing to account for this anomalous observation.

**Dynamics of reversed-intersystem crossing in the strong coupling regime.** So far, an energy inversion between the polaritonic and triplet states in Cavities 3 and 4 has been demonstrated. We will now explore the effect of this inversion on the dynamics of delayed fluorescence. Mechanistically, delayed fluorescence is caused by thermally activated reversed-intersystem crossing from the $T_1$ to $S_1$ state followed by fluorescence. An Arrhenius equation can model the rate of the process [45]:

$$k_{T/S(P^-)} = A \cdot \exp\left(\frac{-E_a}{k_B \cdot T}\right), \qquad (1)$$

where $k_{T/S(P^-)}$ is the RISC rate constant, $k_B$ is the Boltzmann factor, $T$ is the temperature, and $E_a$ is the energy barrier between the triplet and singlet (or $P^-$) states. With increasing temperature, the energy barrier between the triplet and singlet state is easier to overcome. Consequently, the lifetime of the triplet excited state ($\tau_T$) decreases when the temperature rises. The gated temperature-dependent emission spectra and associated emission lifetimes for DABNA-2 films inside and outside cavities are displayed in Fig. 3 (see also Supplementary Table 2). From the neat film, we observed that the lifetime of delayed fluorescence decreased as the temperature increased. However, the intensities of fluorescence first rose and then decreased with increasing temperature. These observations were attributed to the different dependencies that the rates of RISC and non-radiative decay have on temperature. At low temperatures, the variation of $k_{T/S(P^-)}$ plays an more important role than non-radiative processes. Thus, TADF became rapidly intensified with an increasing temperature. However, as the temperature further increased, non-radiative processes became more influential, and the total intensity therefore decreased. For Cavities 1 and 2, similar trends as for the neat

film were observed. As the temperature increased, the intensity of phosphorescence decreased and polariton emission first increased then decreased.

We then turned our attention to the two cavities having inverted energetics of their triplet and $P^-$ states. Cavity 3 displayed negligible differences in the emission envelope from 500 to 550 nm between 77 and 181 K (Fig. 3d), which strengthened the hypothesis of an energy flow from $T_1$ to $P^-$. Furthermore, the intensity of polariton emission from Cavity 3 was ten times more intense at 77 K compared to all other cases, and it decreased steadily as the temperature increased until a plateau value was reached (Fig. 3i). Contrarily, the temperature-dependent emission from Cavity 4 resembled that of the bare film. In other words, Cavity 3 was unique in the series by exhibiting a continuously decreasing polariton emission in the 77–125 K temperature range. Moreover, it should be noted that the weak emission from uncoupled molecules in Cavity 3 followed the same trend as for the neat film (Supplementary Fig. 8). Thus, the dynamical avenues of coupled and uncoupled molecules differed in the same cavity. Furthermore, the lifetime of delayed emission from Cavity 3 decreased as the temperature increased (Fig. 3k–o), but a plateau value was reached at 125 K. As the temperature increased further to around 150 K, the lifetime began to drop again. This indicated the presence of two thermally activated processes working in different temperature regimes in Cavity 3. In the next section, we will analyze the temperature resolved rate of delayed emission to pinpoint the nature of the processes, which caused the differences in delayed emission lifetimes.

The lifetime of emission depends on both the rate of emission and the rates of non-radiative decay. Hence, it is difficult to deduce mechanistic understanding directly from measured lifetimes. To increase our insights on delayed emission in the strong coupling regime, the relative rates of delayed emission were calculated using Eq. 2 (see "Methods" section):

$$k_{T/S(P^-)}(T) \propto \frac{I_{DF}(T)}{I_{PF}(T) \cdot \tau_{DF}(T)}, \qquad (2)$$

where $I_{PF}$ is the integrated prompt emission (Supplementary Fig. 9), $I_{DF}$ is the integrated intensity of delayed emission (Supplementary Fig. 10), and $\tau_{DF}$ is the lifetime of delayed emission. The natural logarithm of the relative rate of delayed emission as a function reciprocal temperature is displayed in Fig. 4a. In the whole temperature range, the relative rates of delayed emission in the neat film, as well as in Cavities 1, 2, and 4, followed exponential growth. This indicated that RISC is a thermally activated process in these systems, and the activation

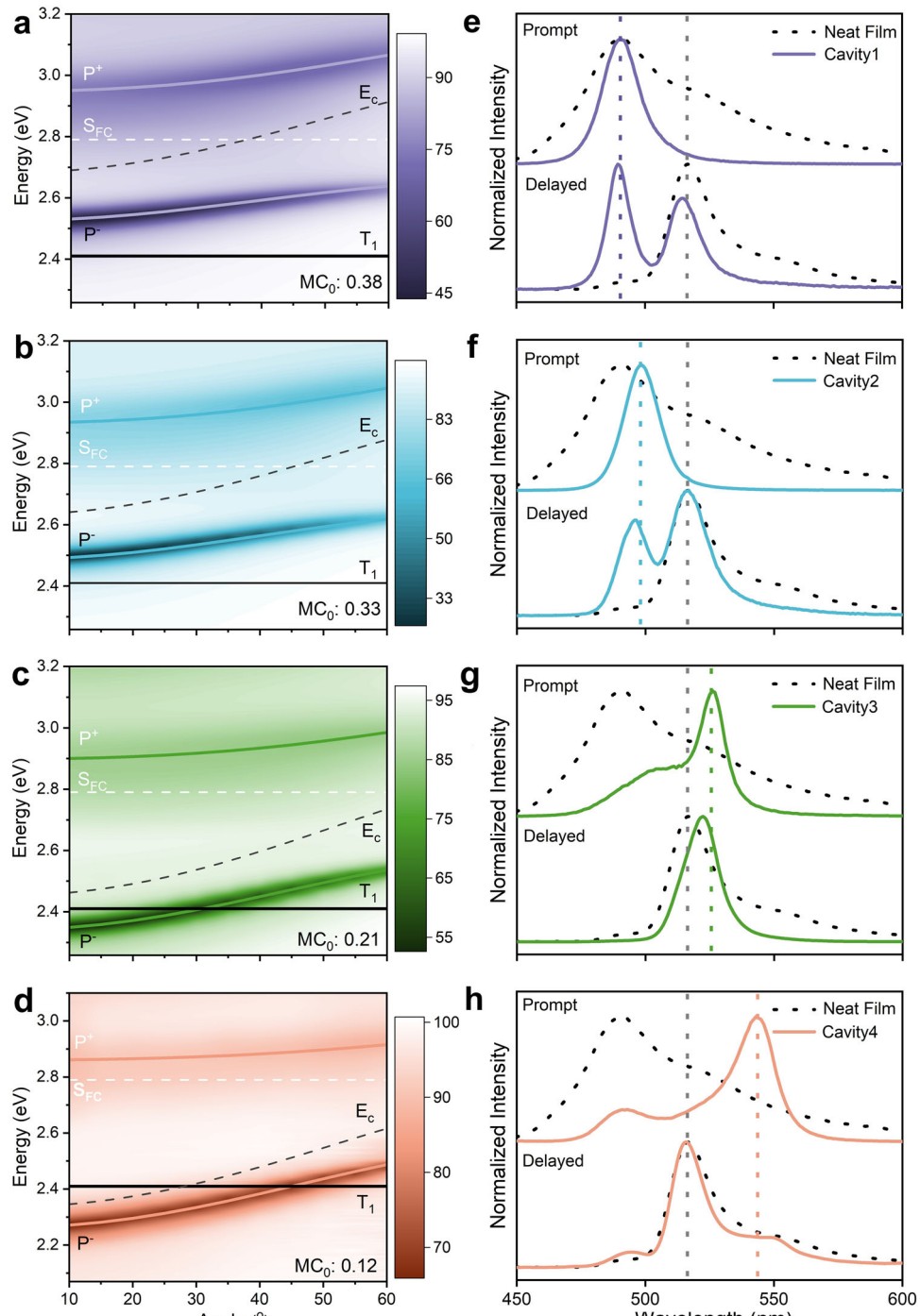

**Fig. 2 Characterization of strong light-matter coupling. a–d** Angle dependent reflectance of Cavity1 (**a**), 2 (**b**), 3 (**c**), and 4 (**d**) with molecular energy levels ($S_{FC}$ = Frank-Condon state, $T_1$ = Triplet state, and $E_c$ = Cavity energy) and fitted polariton dispersions ($P^+$ and $P^-$), $MC_0$ indicate the molecular contribution to $P^-$ at $0°$. **e–h** Prompt (recorded at room temperature) and delayed (recorded at 77 K) emission spectra of a neat film (dashed black line), Cavity1 (**e**), 2 (**f**), 3 (**g**), and 4 (**h**). For Cavities 1–3 the polariton emission maxima moves to higher energies upon cooling. The blue shift is of the same magnitude (approximately 2 nm) and therefore attributed to a small contraction of the film when cooling. The emission maximum of phosphorescence from a neat film is indicated with a vertical gray dashed line in all graphs, and the energy of the lower polariton, as deduced from reflectivity measurements, is indicated with a vertical dashed line of corresponding color.

energies were determined to 115, 96, 65, and 118 meV for the neat film, Cavity 1, Cavity 2, and Cavity 4, respectively (Eq. 1). The activation energies of RISC in Cavities 1 and 2 were nearly 20 and 50 meV smaller than for the neat film, which imply a direct pathway from the triplet state to $P^-$. For Cavity 4, the effect of singlet–triplet inversion was negligible for the dynamics of

delayed emission, which indicated that the process is dominated by transfer from the triplet to excitonic states followed by partial relaxation to the lower polariton. Interestingly, Cavity 4 show a significant amount of delayed excitonic emission at 180 K (Fig. 3e). Energy transfer from both the singlet and triplet surfaces to $P^-$ is therefore inefficient in this cavity. We speculate

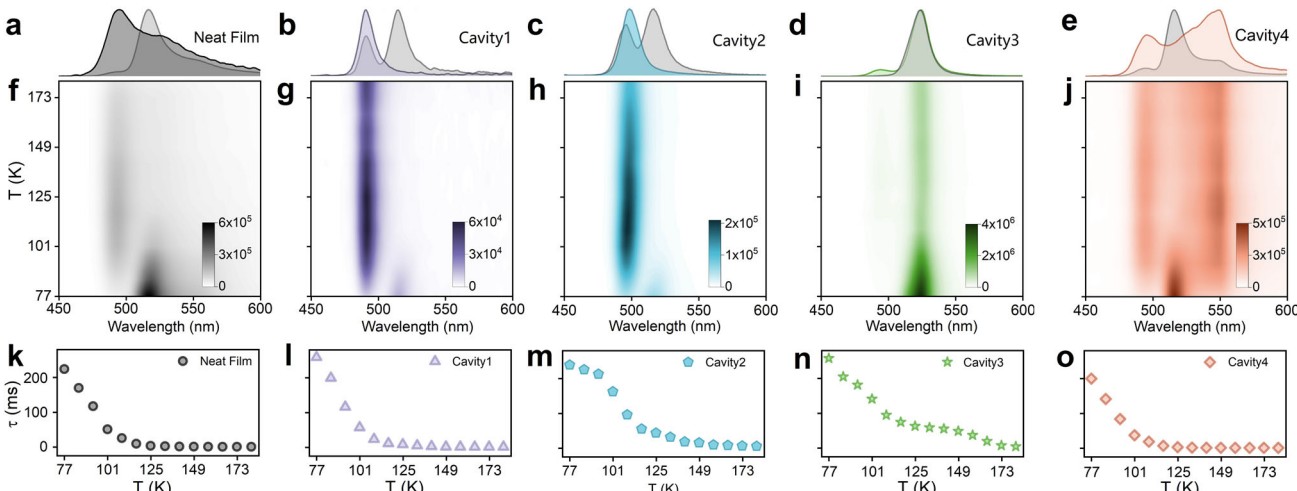

**Fig. 3 Temperature dependent steady state and time resolved delayed emission. a–e** Normalized gated emission (gate delay 100 ns) at 181 K (colored shaded areas) and 77 K (gray shaded areas). **f–j** Temperature-dependent gated emission (gate delay 100 ns). The emission from Cavity 3 is approximately an order of magnitude larger than for the others. **k–o**, Temperature-dependent lifetimes of delayed emission.

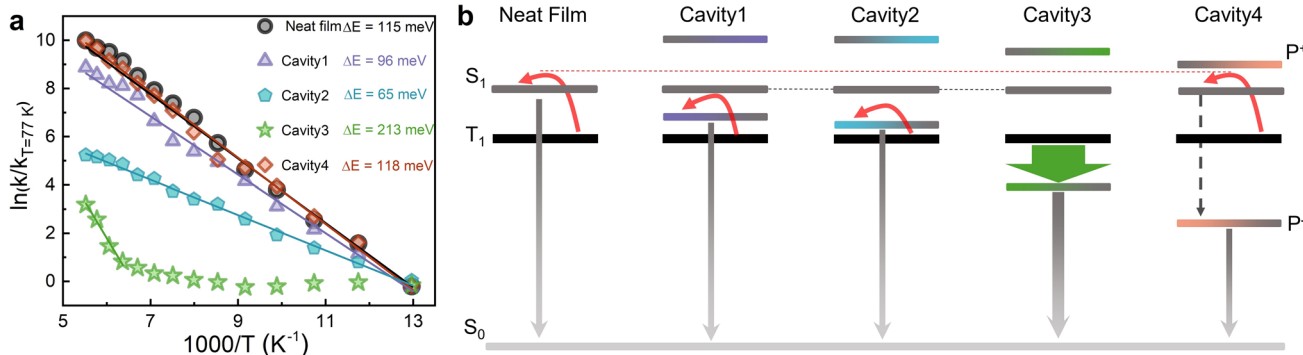

**Fig. 4 Barrier-free RISC by strong coupling. a** Arrhenius analysis with calculated activation energies for delayed P⁻ (or singlet) emission inside and outside cavities. Lines show best fit to Eq. 1. **b** Schematic illustration of the dynamics of the triplet to polariton (or singlet) transition at low temperatures. Red arrows represent thermally activated and green represents barrier-free RISC. The dashed black arrow indicates a relaxation from $S_1$ to $P^-$, and the gray one represents delayed emission from $S_1$ or $P^-$.

that the low coupling to the lower polaritonic state is due to the large red-detuning of the cavity, which in such case would rationalize recent observations by Eizner et al. [37].

In contrast, the dynamical avenue in Cavity 3, the other states-inverted cavity, was different. From 77 to 150 K, the derivative of the logarithmic rate of delayed emission with respect to reciprocal temperature was zero. The rate of delayed emission in Cavity3 was therefore not thermally activated, and a barrier-free and exothermic conversion from the triplet to the polaritonic state was achieved. However, when temperature was further increased, another pathway between the triplet and polaritonic states started to play a role. This new pathway was thermally activated with an activation energy of 213 meV, which was considerable larger than that of the neat film. We cannot say if the observed thermally activated process was due to direct or bare exciton mediated transfer to P⁻. Nevertheless, the high activation energy indicated large perturbations to the excited state surface. Furthermore, the fact that the process was observable at all suggested that the coupling for the thermally activated phenomena was larger than for the barrier-free one, which presently limits the phenomena to low temperatures.

Both polariton energetics and composition must be considered in order to explain the dynamics of RISC for DABNA-2 in the strong coupling regime (Fig. 4b). With a close to tuned system

(Cavities 1 and 2), P⁻ had a considerable molecular component, and direct energy transfer from the triplet states to P⁻ was evident from the reduced activation barriers. By detuning the cavity to lower energies, the P⁻ energy was below $T_1$ (Cavity 3). The emission intensified and the rate of RISC was temperature independent in the low temperature range. When red tuning was further increased (Cavity 4), the apparent driving force for RISC was expected to increase. However, the connection between the triplet and polaritonic states disappeared, and the energy from $T_1$ first transferred to uncoupled singlet excited states followed by partial relaxation to P⁻. We speculate that the reason for this observation is the low molecular contribution to P⁻, which was less than fifteen percent (Supplementary Fig. 6). This observation highlights the importance of polariton constitution, when assessing the interplay between hybrid light matter and molecular centered states.

The analysis so far has assumed that the energy of the triplet state is not greatly affected by the presence of a close to resonant cavity. To confirm the validity of this assumption, angular resolved reflectivity, prompt emission, and delayed emission was recorded on a cavity having similar parameters as Cavity 3 (Fig. 5, and Supplementary Table 3). The energy of the dispersive P⁻ state was extracted from the angular resolved reflectivity, and it matched the energy of prompt emission. Delayed emission at

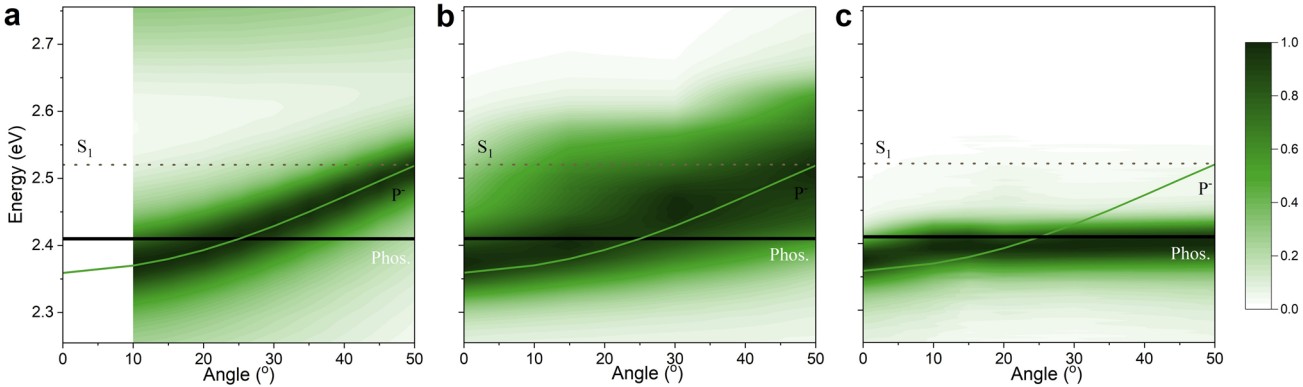

**Fig. 5 The crossing point between polariton emission and phosphorescence. a–c** Angle dependent absorption (1-reflectance) (**a**), angle dependent prompt emission at room temperature (**b**), and angle dependent delayed emission at 77 K (**c**) of a cavity having similar parameters as Cavity 3. Fluorescence (dotted lines), phosphorescence (black solid lines), and fitted P⁻ energy (green solid lines) are indicated. It should be noted that the excitation angle could not be held constant when performing angular dependent delayed emission experiments at 77 K. The emission dispersion plot in **c** is therefore normalized, with the consequence that information on the relative intensity of emission as a function of angle is lost.

77 K can occur directly from the non-dispersive triplet state or as a result of a transition from the triplet to the polaritonic state followed by P⁻ emission. At a low angle of incidence, the energy of delayed emission matches well with the energy of P⁻ (as deduced from the angular resolved reflectivity measurements). However, the P⁻ energy increases with the angle of incidence but the emission energy from this cavity increases in energy only up to a certain point, which coincides with the energy of the triplet state (as deduced from the phosphorescence maximum from neat film experiments). The angular resolved emission thus follows the expected crossing point in energy between the P⁻ and triplet states. This experiment shows that the effect by the cavity on the triplet state energy is negligible even at close to resonant conditions, a result most likely due to the very low transition dipole moment of the triplet state.

## Discussion

We demonstrate a barrier-free transition from a molecular centered triplet state and a hybrid light matter state. By sweeping the optical resonance of a Fabry–Perot cavity containing an organic dye, polaritonic emission from higher to lower energies as compared to molecular phosphorescence was achieved. The connection between the uncoupled triplet state and the polaritonic states was shown to depend on the composition of the polariton. As the photonic nature of P⁻ increases, a gradual disconnection from the triplet state occurs. We hypothesize that the benefit of the driving force given by a large cavity detuning is lost due to the lower molecular constitution of the polariton. At an intermediate detuning, a system was created with both an energetic driving force and enough material character of the lower polariton to sustain a barrier-free reversed-intersystem crossing directly from the triplet to the polaritonic state. Accordingly, strong light-matter coupling provides a new strategy to break the limit of Hund's rule and facilitate triplet energy harvesting. From these results, light-matter interaction between molecules and cavities is anticipated to become more deeply explored for OLED and organic laser applications [46].

## Methods

**Cavity fabrication.** The Fabry–Pérot cavities were built on glass substrates (25 × 25 mm²), which were pre-cleaned by sonication for 15 min in alkaline solution (0.5% of Hellmanex in distilled water), and then rinsed with water and sonicated for 1 h in water and ethanol, respectively. The cleaned glass substrates were dried in an oven overnight before cavity preparation. Molecules were dissolved in toluene (or chloroform), and then deposited by spin-coating (Laurell Technologies WS-650), at speeds from 1200 to 2000 rpm to give roughly 120–150 nm thick films. The

thickness of the film was optimized to strongly couple the λ/2 cavity mode with the molecular transition. Ag mirrors were fabricated by vacuum sputtering deposition (HEX, Korvus Technologies). A thick 100 nm Ag film was sputtered on top of the glass plate and the semitransparent 40 nm film was sputtered on top of the molecular film to complete the cavities.

**Optical measurements.** Steady-state absorption spectra were measured using a standard spectrophotometer (Lambda 950, Perkin Elmer). The angle-resolved reflectivity was measured using the same spectrophotometer equipped with the Universal Reflectance Accessory (URA)—LAMBDA. Prompt emission measurements were performed using an FLS 1000 spectrofluorometer (Edinburgh Instruments). The angle-dependent steady-state emission were measured using a homemade goniometer consisting of two liquid light guides connected to the spectrofluorometer. The light from the fiber guiding the excitation light was focused on the sample at a fixed angle of 30°, and the emitted light was focused on the entrance of the second fiber, which was placed at different angles and was used to guide the light back to the detector of the spectrofluorometer. Prompt emission lifetime measurements were determined with a time-correlated single-photon counting system (FLS1000) using picosecond-pulsed diode lasers. The frequency of the pulsed laser was set to 2 MHz, with a pulse duration of 100 ps. Delayed emission spectra were performed on an Edinburgh Instrument LP 980 spectrometer equipped with an ICCD (Andor). A Spectra-Physics Nd:YAG 532 nm laser (pulse width ~7 ns) coupled with a Spectra-Physics primoscan optical parametric oscillator (OPO) was used as pump source. RISC dynamics were measured using the same laser with the emission recorded by an FLS1000 in MCS mode. The lifetime of polariton emission has been shown to be independent of angle incidence of the emitted light[47], which was confirmed here by observing the same emission lifetime when recording the delayed emission from Cavity 4 at 10 and 40 degrees (Supplementary Fig. 11). All dynamics presented elsewhere in the manuscript and supplementary information were measured at normal angle of incidence. Temperature dependent decays are shown in Supplementary Fig. 12.

**Elaboration of the molecule screening.** To reach strong coupling regime, the concentration of TADF molecules in the form of films should be relatively high. Meanwhile, fluorescence is often quenched at high concentrations. The lifetime of emission at different dye/polymer concentrations was measured to assess the concentration quenching effect. Supplementary Fig. 2 shows the lifetimes of the five TADF organic films at different dye concentrations in a polymer film. Taking their high initial emission quantum yield into account, the concentration quenching effect won't hinder optical characterization when reaching the strong coupling regime of any of the five compounds.

Energy inversion will require molecules having a small Stokes shift and large Rabi splitting ($\hbar\Omega_R$). Supplementary Fig. 3 shows absorbance and emission from molecular films of the five compounds. DABNA-1, DABNA-2, and TBN-TPA have smaller Stokes shift than 3DPyM-pDTC and the boron difluoride curcuminoid derivative. Angle resolved reflectivity of Fabry–Pérot cavities containing neat films of these three derivatives, where therefore made as to assess their possibility to enter the strong coupling regime (Supplementary Fig. 4). DABNA-2 has the largest Rabi splitting of the three derivatives (Supplementary Table 1). Furthermore, phosphorescence from DABNA-2 films at cryogenic temperatures can be probed, allowing for accurate determination of the energy of the triplet state, and this molecule was therefore chosen in the study. Finally, prompt emission from Fabry–Pérot cavities containing DABNA-2 show a dispersive behavior (see Supplementary Fig. 13).

**Coupled oscillator model.** To extract the coupling strength, polariton energies, and polariton composition, the dispersion of polariton energies were fitted to a coupled harmonic oscillator model:

$$\begin{pmatrix} E_c(\theta) & \frac{\hbar\Omega_R}{2} \\ \frac{\hbar\Omega}{2} & E_x \end{pmatrix} \begin{pmatrix} \alpha \\ \beta \end{pmatrix} = E(\theta) \begin{pmatrix} \alpha \\ \beta \end{pmatrix} \tag{3}$$

where $E_c$ is the cavity photon energy, which is related to the incident angle $\theta$, $E_x$ is the exciton energy (the Frank-Condon state), and $\hbar\Omega_R$ is the Rabi splitting. The in-plane distribution of polariton energies are obtained from the eigenvalues of the Hamiltonian. The Hopfield coefficients, $|\alpha|^2$ and $|\beta|^2$, represent the fractional excitonic and photonic contributions to the corresponding polaritons.

**Analysis of $k_{T/S(P^-)}$ as the temperature changes.** The RISC rate constant ($k$) can be calculated from experimental observables using the following equation [4,37,48]:

$$k_{T/S(P^-)} = \frac{k_{PF}k_{DF}\Phi_{DF}}{k_{ISC}\Phi_{PF}}, \tag{4}$$

where $k_{PF}$ and $k_{DF}$ are the prompt and delayed fluorescence rate constants, respectively, $\Phi_{PF}$ and $\Phi_{DF}$ are the quantum yields of the prompt and delayed components, and $k_{ISC}$ is the intersystem crossing rate constant. As the temperature changes, $k_{PF}$ and $k_{ISC}$ stay constant. Therefore, the following relationship can be used:

$$k_{T/S(P^-)} \propto \frac{k_{DF}(T) \cdot \Phi_{DF}(T)}{\Phi_{PF}(T)} \tag{5}$$

Meanwhile the quantum yields are proportional to the integrated emission intensities, in addition $k_{DF}$ and $\tau_{DF}$ follow the relationship below:

$$\Phi_{DF}(T) \propto I_{DF}(T) \tag{6}$$

$$\Phi_{PF}(T) \propto I_{PF}(T) \tag{7}$$

$$k_{DF} = \frac{1}{\tau_{DF}} \tag{8}$$

Inserting Eqs. (5–8) into (4), gives Eq. (2). It should be noted that the reason for using the proportionality between the intensity of emission and the emission quantum yields, is that emission from polaritonic states are dispersive and must therefore be measured in an integrating sphere. Ours is not compatible with cryogenic temperatures, with the solution to this dilemma being that relative rates is displayed in Fig. 3a. This unable comparison of rates between samples, but captures the temperature dependence, and thus enables the calculation of energy barriers.

## Data availability

Source data are provided with this paper

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

## Acknowledgements

We gratefully acknowledge financial support from the European Research council (ERC2017-StG-757733) and the Knut and Alice Wallenberg Foundation (KAW 2017.0192). Joel Yuen-Zhou is acknowledge for discussions on theory.

## Author contributions

K.B. conceived the project and supervised the work. S.M. carried out the screening and selection of potential molecules and their synthesis. Y.Y. prepared the solid film and cavity samples, performed the spectroscopy measurements, analyzed spectroscopic data, and carried out the corresponding simulations with the help from M.W. All authors contributed to the writing and editing of the manuscript and have given approval to its final version.

## Funding

## Competing interests

The authors declare no competing interests.
