## [Peer Review File · Nature Communications]

Reviewers' Comments:

Reviewer #1:

Remarks to the Author:

The manuscript reports a set of experimental results on temperature and angle-resolved emission of a strongly coupled organic microcavity with metallic mirrors, where the inversion of singlet and triplet transitions induced by the strong coupling of molecular excitations to the cavity modes might take place. The singlet-triplet inversion is a remarkable effect hugely important for the realization of efficient organic LEDs and lasers. Its experimental observation would certainly deserve publication in Nature Communications. The conclusion on the observation of the desired effect is based on the analysis of temperature dependences of intensities and life-times of the luminescence peaks. The angular dependence of the peaks certifies of a large Rabi-splitting of polariton modes. In some cases the lower polariton mode goes below the triplet state that allows for the inversion, potentially. However, no features corresponding to the presumable singlet-triplet crossing points are seen in the angle-dependent spectra. Thus the question arises: is the triplet transition really situated at the energy where the authors place it? As the optical transitions in this system are subject to a very strong inhomogeneous broadening, this question becomes challenging, from my opinion. The interpretation of the data is based on the oversimplified two-coupled oscillator model that ignores the inhomogeneous broadening and on some phenomenological relations. No attempt to simulate the relaxation dynamics of polaritons with, say, Boltzmann kinetic equations is shown. In conclusion, the data would deserve publication if accompanied by either an unambiguous direct experimental evidence for singlet-triplet crossing or a comprehensive theoretical model that would shed light on a complex relaxation processes in the system and confirm the interpretation of the data proposed in the manuscript.

Reviewer #2:

Remarks to the Author:

This manuscript is a follow-up investigation of the authors' previous work exploring the extent to which strong coupling can alter the dynamics of RISC by creating an exothermic pathway from the triplet to the lower polariton. By varying the cavity detuning, the authors show that there is a sweet spot in which the exothermic condition is achieved and the polariton mode is sufficiently matter-like to effect transfer from the triplet directly into the lower polariton. The work is nicely presented and technically well-done. The topic is a fascinating one and, though many questions remain, this work will be a useful contribution for helping the field move forward. I have a few questions/suggestions below that may help strengthen the manuscript.

1. The authors interpret the peak at ~ 520 nm as phosphorescence but it wasn't clear how they distinguish this peak from delayed fluorescence that just has a different weighting of the vibronics (i.e. peaking at the 0-1 transition rather than peaking with the 0-0 transition in the case of the prompt fluorescence)? It seems like a remarkable coincidence that the peak of the phosphorescence would coincide exactly with the 0-1 shoulder of the prompt fluorescence.

2. The authors hypothesize that the reason cavity 4 doesn't show a significant effect is because the polariton is too photon-like at such negative detuning for the transfer to occur. This explanation seems tenuous for several reasons:

a) The Hopfield coefficient of cavity 4 is only a factor of two different at normal incidence (and less at higher angles) from cavity 3. This is a pretty minor difference to have such a big effect.

b) Cavity 4 achieves the same polariton energy and Hopfield coefficient at $\sim 40^\circ$ as cavity 3 does at 10° . Since the triplets presumably have the ability to transfer to a polariton at any wave vector (and the density of polariton states increases with angle), why wouldn't cavity 4 achieve the same exothermic transfer that cavity 3 does? If it did, the polaritons would either relax down the branch toward $k=0$ (in which case the authors would have seen similar behavior to cavity 3) or the polaritons decay before relaxing, in which case I would expect to see altered kinetics when measuring off angle. Does the DF transient of cavity 4 change between 10° and 40° ? In general, I

think it would be really useful to provide the PL intensity dispersion map, not just for the steady-state emission as in Fig. 11, but for the prompt and delayed emission as well as a justification for why the authors choose to focus their interpretation based on $k \sim 0$ emission as opposed to elsewhere in the dispersion.

4. The prompt fluorescence lifetime of DABNA-2 in Fig. S2c (I assume that one is DABNA-2) depends strongly on concentration in polystyrene. Does this also affect its DF transient? If intermolecular interactions have such a big effect on the PL kinetics, might those rates also get affected by strong coupling and be another piece to the puzzle?

5. The quantum yields in Eq. 4 & 5 of the methods section are missing.

Reviewer #3:

Remarks to the Author:

The paper reports an interesting application of cavity-controlled photo-physics, namely, the ability to reverse the usual energetic ordering of triplet and singlet states for organic molecules inserted into a tuned microcavity, thereby leading to barrierless reverse intersystem crossing, RISC. The authors present some rather convincing evidence of the effect for an organic chromophore DABNA-2: When the Rabi frequency is sufficiently large the lower polariton becomes positioned below the triplet state leading to an intense, delayed emission which monotonically decreases with temperature, reflecting the lack of an energy barrier that normally exists in RISC from neat films. However, if the lower polariton is detuned too far below the triplet, the barrierless RISC effectively shuts off, an effect the authors attribute to the lack of a substantial material component in the lower polariton. The paper is well-written, logically developed with solid, supportive evidence to back their hypothesis. I recommend publication after the following minor points are addressed.

1) I could not find in the text where the lifetime of the prompt emission in DABNA-2 neat films is reported. This is important because the delayed emission was always measured after waiting 100 nsec. I assume that 100 nsec is much greater than the fluorescence lifetime? The authors should make a point of this to ensure that the delayed fluorescence is entirely derived from RISC.

2) It would be very helpful if the phosphorescence peak in figure 2 was labelled and perhaps marked with a vertical line. The lower polariton emission peak should also be labelled.

3) The authors need to better justify the cavity parameters. Importantly, why does cavity 4 have 50% larger Rabi frequency than cavity 1??? The Rabi frequency should scale as the sqrt of the molecular density. Is cavity 4 more dense?

4) There should be some more discussion as to why the lower polariton in cavity 4 has a smaller material component than the polariton in cavity 3 despite cavity 4 having a larger Rabi frequency.

Reviewer #1:

The manuscript reports a set of experimental results on temperature and angle-resolved emission of a strongly coupled organic microcavity with metallic mirrors, where the inversion of singlet and triplet transitions induced by the strong coupling of molecular excitations to the cavity modes might take place. The singlet-triplet inversion is a remarkable effect hugely important for the realization of efficient organic LEDs and lasers. Its experimental observation would certainly deserve publication in Nature Communications. The conclusion on the observation of the desired effect is based on the analysis of temperature dependences of intensities and life-times of the luminescence peaks. The angular dependence of the peaks certifies of a large Rabi-splitting of polariton modes. In some cases the lower polariton mode goes below the triplet state that allows for the inversion, potentially. However, no features corresponding to the presumable singlet-triplet crossing points are seen in the angle-dependent spectra. Thus the question arises: is the triplet transition really situated at the energy where the authors place it? As the optical transitions in this system are subject to a very strong inhomogeneous broadening, this question becomes challenging, from my opinion. The interpretation of the data is based on the oversimplified two-coupled oscillator model that ignores the inhomogeneous broadening and on some phenomenological relations. No attempt to simulate *the relaxation dynamics of polaritons* with, say, Boltzmann kinetic equations is shown.

In conclusion, the data would deserve publication if accompanied by either an unambiguous direct experimental evidence for singlet-triplet crossing or a comprehensive theoretical model that would shed light on a complex relaxation processes in the system and confirm the interpretation of the data proposed in the manuscript.

We would like to thank the reviewer for highlighting the importance of firmly concluding that the energy of the triplet state is unperturbed inside the cavities in our study. We have tackled this challenge as follows: A new cavity was made, having similar parameters to Cavity3 (Cavity3 has degraded and can therefore not be used more). Then we performed angular resolved reflectivity, angular resolved prompt emission (at room temp), and angular resolved delayed emission (at 77K), on this new cavity. From the angular resolved reflectivity, we could get the polariton energy as a function of angle of incidence. From the angular resolved prompt emission (That only prompt emission is recorded in the experiment, excludes all involvement of triplet states), we could show that emission from the system follows the polariton energy. From the angular resolved delayed emission (That only delayed emission is recorded in the experiment, ensures that the triplet state must play a role in the photophysical pathways), we could then study any eventual crossing point between polaritonic emission and phosphorescence. New Figure 5 (see below) show the results from these experiments, with the “c” part showing the angular resolved delayed emission. At a low angle of incidence, the energy of the emission coincides with the polariton energy. When the angle of emission is increased, the energy of emission is first increased (following the increase in the energy of the polariton), but at higher angles the energy of emission is constant with angle. This in contradiction to the prompt emission experiment (part “b” in the figure below), in which the polariton emission energy constantly increases with angle. Furthermore, the energy plateau seen in the “c” part coincides with the energy of phosphorescence. We therefore suggest that at low emission angles, the emission is polaritonic and at high emission angles, the emission occurs through phosphorescence. These

data suggests that a singlet-triplet crossing point occurs at the intersect between the polaritonic and triplet energies. That this crossing point can be seen in the angular resolved delayed emission spectra is indicative that we have placed the triplet energy at the correct position in our analysis. The updated manuscript contains a new section discussing this experiment, and also a new figure (Figure 5). The new section reads:

“The analysis so far has assumed that the energy of the triplet state is not greatly affected by the presence of a close to resonant cavity. To confirm the validity of this assumption, angular resolved reflectivity, prompt emission, and delayed emission was recorded on a cavity having similar parameters as Cavity3 (Figure 5, Supplementary Table 3). The energy of the dispersive P- state was extracted from the angular resolved reflectivity, and it matched the energy of prompt emission. Delayed emission at 77 K can occur directly from the non-dispersive triplet state or as a result of a transition from the triplet to the polaritonic state followed by P- emission. At a low angle of incidence, the energy of delayed emission matches well with the energy of P- (as deduced from the angular resolved reflectivity measurements). However, the P- energy increases with the angle of incidence but the emission energy from this cavity increases in energy only up to a certain point, which coincides with the energy of the triplet state (as deduced from the phosphorescence maximum from neat film experiments). The angular resolved emission thus follows the expected crossing point in energy between the P- and triplet states. This experiment shows that the effect by the cavity on the triplet state energy is negligible even at close to resonant conditions, a result most likely due to the very low transition dipole moment of the triplet state.”

Fig. 5 The crossing point between polariton emission and phosphorescence. a-c, Angle dependent reflectance (a), angle dependent prompt emission at room temperature (b), and angle dependent delayed emission at 77 K (c) of a cavity having similar parameters as Cavity3. Florescence (dotted lines), phosphorescence (solid lines) and fitted P⁻ energy (dashed lines) are indicated. It should be noted that the excitation angle could not be held constant when performing angular dependent delayed emission experiments at 77 K. The emission dispersion plot in (c) is therefore normalized, with the consequence that information on the relative intensity of emission as a function of angle is lost.

Developing a comprehensive theoretical model is quite far beyond our capability as experimentalists. However, we would like to provide a heuristic theoretical view of what we think happens in Cavity 3. Two processes are competing rate wise against each other: The triplet to polaritonic transition and the triplet to singlet transition. The triplet state is localized,

whereas the polariton is delocalized. A triplet to polariton type of transition is often assumed to have a reduced probability due to a low wave function overlap between these two states. The reduced wave function overlap is expected to slow down this transition by $1/N$, where N is the number of molecules coupled to the cavity mode (often assumed to be on the order of $10^5 - 10^6$). Let us assume then that in Cavity3 (which has the triplet and polaritonic states energetically inverted), the rate of triplet to polaritonic transfer is $1/N$ slower as compared to the rate of inter system crossing. Outside cavities, there is no penalty for a reduced wave function overlap; however, there is an energetic barrier for the triplet to singlet transition. Let us assume that this follows an Arrhenius behavior with a pre-exponential factor equal to the rate of inter system crossing. The result of this argumentation are shown in the figure below, together with the data of Cavity3. Cavity3 shows a transition from a barrier-free reversed intersystem crossing, to an intersystem crossing containing a barrier at around 150 K. In the manuscript, we speculate that this occurs due to a mechanism change from a triplet to polaritonic transition at low temperatures, to a triplet to singlet transition at higher temperatures. This can be reproduced reasonably well using the heuristic model described above, using the literature value of the energy difference between the triplet and singlet energies and an efficient delocalization of the polariton over 10^5 molecules.

Having said this, we note that recent theoretical predictions have shown that the ultra-fast decay of the intra-cavity photon may play a significant role in facilitating cavity mediated photochemical processes (J. Chem. Phys. 2020, 153 (23), 234304; J. Phys. Chem. Lett. 2020, 11 (20), 8810-8818; Chem 2020, 6 (1), 250-265; J. Chem. Phys. 2021, 154 (5), 054104). Even though the overlap of a single triplet state wave function with the collective lower polariton state may be small, the population in the lower polaritonic state is removed on a sub 100 fs time scale by means of photon decay. It can thus be assumed that the back-reaction, becomes sufficiently suppressed, counteracting the low wave function overlap.

Figure caption (top) Blue, red and black lines show the effect of reversed intersystem crossing rate constant due to the energy barrier in the conversion in the neat film (e.i. $\exp(-\Delta E/k_B \cdot T)$). Green and purple lines show the effect of the reversed intersystem crossing rate constant due to a reduced wave function overlap in the cavity systems, using often assumed values for how many molecules that are coupled to a cavity mode.

Reviewer #2:

This manuscript is a follow-up investigation of the authors' previous work exploring the extent to which strong coupling can alter the dynamics of RISC by creating an exothermic pathway from the triplet to the lower polariton. By varying the cavity detuning, the authors show that there is a sweet spot in which the exothermic condition is achieved and the polariton mode is sufficiently matter-like to effect transfer from the triplet directly into the lower polariton. The work is nicely presented and technically well-done. The topic is a fascinating one and, though many questions remain, this work will be a useful contribution for helping the field move forward. I have a few questions/suggestions below that may help strengthen the manuscript.

1. The authors interpret the peak at ~520 nm as phosphorescence but it wasn't clear how they distinguish this peak from delayed fluorescence that just has a different weighting of the vibronics (i.e. peaking at the 0-1 transition rather than peaking with the 0-0 transition in the case of the prompt fluorescence)? It seems like a remarkable coincidence that the peak of the phosphorescence would coincide exactly with the 0-1 shoulder of the prompt fluorescence. The reviewer is right that the phosphorescence at 515 nm overlaps with the 0-1 shoulder of fluorescence. We would expect this to be a common feature of most TADF molecule exhibiting phosphorescence, because the triplet-singlet energy difference in this class of molecules is in the 20-200 meV range. To distinguish phosphorescence from fluorescence, temperature dependent spectroscopy was performed. At 77K we see mostly emission at 515 nm in the bare film. However, as the temperature is increasing, the intensity of this peak decreases, and another peak is appearing (Figure 3f). This new peak has the same spectral envelope as prompt fluorescence. This together with the long-lived and oxygen sensitive of the state responsible enables us to deduce that it is phosphorescence. We should also note that this molecule is known, and other groups have come to the same conclusion. We have rewritten the section that introduce the photophysics of DABNA-2 to clarify how to distinguish fluorescence and phosphorescence. This section now reads:

“Gated emission at 77 K showed weak residual fluorescence as well as phosphorescence at longer wavelengths (~515 nm). The long-lived fluorescence is a result of RISC from the excited triplet to singlet states by thermal activation. These two processes could therefore be distinguished from each other by observing their temperature dependencies (vide infra). The ΔE value derived from these measurements was 123 meV, which was consistent with previous studies of 1 wt% films (140 meV).⁴⁰ We will later show that the small delayed fluorescence shifted to lower energies and was therefore greatly enhanced when the molecule was strongly coupled to an optical cavity.”

2. The authors hypothesize that the reason cavity 4 doesn't show a significant effect is because the polariton is too photon-like at such negative detuning for the transfer to occur. This explanation seems tenuous for several reasons:

a) The Hopfield coefficient of cavity 4 is only a factor of two different at normal incidence

(and less at higher angles) from cavity 3. This is a pretty minor difference to have such a big effect.

We agree with the reviewer that the change in Hopfield coefficients could be viewed as small. But it is not only the delayed emission where we see a disconnection between the molecular centered and polaritonic states. When we probe the emission, we see more emission that is excitonic from this cavity as compared to others. This indicates that the scattering between the exciton reservoir and the lower polariton is less efficient in this cavity. This observation goes hand in hand with the observation of an inefficient scattering from the triplet to the polaritonic state. In the updated manuscript, we draw the parallel of inefficient scattering from both the S1/T1 states to the lower polariton. We also moderate our discussion on the reason behind this observation as to indicate that our suggestion is a hypothesis. The updated section reads

“For Cavity4, the effect of singlet-triplet inversion was negligible for the dynamics of delayed emission, which indicated that the process is dominated by transfer from the triplet to excitonic states followed by partial relaxation to the lower polariton. Interestingly, Cavity4 show a significant amount of delayed excitonic emission at 180 K (Figure 3e). Energy transfer from both the singlet and triplet surfaces to P- is therefore inefficient in this cavity. We speculate that the low coupling to the lower polaritonic state is due to the large red-detuning of the cavity, which in such case would rationalize recent observations by Eizner et al.37”

b) Cavity 4 achieves the same polariton energy and Hopfield coefficient at $\sim 40^\circ$ as cavity 3 does at 10° . Since *the triplets presumably have the ability to transfer to a polariton at any wave vector (and the density of polariton states increases with angle)*, why wouldn't cavity 4 achieve the same exothermic transfer that cavity 3 does? If it did, the polaritons would either relax down the branch toward $k=0$ (in which case the authors would have seen similar behavior to cavity 3) or the polaritons decay before relaxing, in which case I would expect to see altered kinetics when measuring off angle. Does the DF transient of cavity 4 change between 10° and 40° ? *In general, I think it would be really useful to provide the PL intensity dispersion map, not just for the steady-state emission as in Fig. 11, but for the prompt and delayed emission as well as a justification for why the authors choose to focus their interpretation based on $k\sim 0$ emission as opposed to elsewhere in the dispersion.*

This is a similar question to the one raised by reviewer 1. A delayed PL dispersion map has now been made. See reply to Reviewer 1 for an elaborated discussion on the conclusions from this new experiment.

We have now measured the lifetime of delayed emission from Cavity 4 at 10 and 40 degrees. The lifetime is the same (see new Supplementary Figure 11 shown below). This result is as expected because the measured lifetime reflects the lifetime of the triplet state. Actually, it should not matter whether the rate of scattering from the triplet state to the polaritonic state is angle dependent, the measured lifetime reflects the combined rate of all processes deactivation the triplet state. It should be noted here that the rate of diffusion of triplet energy between molecules (e.i. exciton hopping between molecules) is larger as compared to the triplet lifetime. This means that the triplet exciton can be regarded as rotating on the triplet

lifetime timescale. The consequence of this is that the fastest scattering angle will be dominating all angles. Furthermore, the lifetime of these systems is to a high extent dominated by vibrational relaxation (e.i. ISC from T1 to S0 followed by VR). This makes emission intensity normalization of the measured lifetimes important, as it results in a rate of emission (This is how we present the data in Figure 4a). A comment highlighting the angle independent emission lifetime is added in the methods section, it reads:

“The lifetime of polariton emission has been shown to be independent of angle incidence of the emitted light,⁴⁷ which was confirmed here by observing the same emission lifetime when recording the delayed emission from Cavity4 at 10 and 40 degrees (Supplementary Fig. 11). All dynamics presented elsewhere in the manuscript and supplementary information were measured at normal angle of incidence.”

Supplementary Figure 11 Time resolved delayed emission of Cavity4, recorded at 10 and 40 degrees.

Regarding why we focus on $k \sim 0$ emission: The most important reason for measuring at $k=0$ is that polariton emission is most intense, and we therefore get the largest signal to noise ratio at this configuration. At higher angles we further risk getting contamination from excitonic emission as the intensity of exciton emission is less angle dependent. We actually see very little prompt polariton emission at 40 degrees in Cavity4 (Figure S13d). This is as expected as the intensity of emission from the lower polariton always (or at least most often) reduces with angle, while the intensity of excitonic emission is much less sensitive on the angle. Furthermore, we do these measurements in a cryostat at 77K. We can turn our sample in the cryostat (we did this in the new Figure 5, see answer to Reviewer 1), but errors in setting this angle is quite large, except at zero degrees. So by constantly measuring at $k=0$, we can attain the highest signal to noise ratio, in combination with the highest correctness of angle settings. We see no obvious drawback of using this configuration.

4. The prompt fluorescence lifetime of DABNA-2 in Fig. S2c (I assume that one is DABNA-2) depends strongly on concentration in polystyrene. Does this also affect its DF transient? If

intermolecular interactions have such a big effect on the PL kinetics, might those rates also get affected by strong coupling and be another piece to the puzzle?

The reviewer is right that the concentration will affect the DF transient. Excited state quenching at high dye concentrations in a film is a well known phenomena. This effect will add as an effective increase in the non-radiative relaxation rate. However, because we compare our cavity experiments with a bare film of the same concentration, the sample and reference has the same non-radiative relaxation rates. Furthermore, and more importantly, since we compare rates of emission rather than lifetimes, non-radiative rates cancels out in the final equation we use (QY of emission/lifetime of emission = rate of emission). Although we do not think it will affect the conclusions in this manuscript, the effect of the strong coupling regime on non-radiative rates is an interesting topic. We cannot remember reading about it. However, there is a recent paper discussion this phenomenon in a J-aggregate. The discussion there was that since the excited state is delocalized, vibrational-relaxation in which all excitation energy goes to a single molecules, is suppressed. This by a reduced wave function overlap argument. The same argumentation could be used in the strong exciton-photon coupling regime, and if this effect could be utilized it could result in a significantly increased emission quantum yield from a strongly coupled system.

5. The quantum yields in Eq. 4 & 5 of the methods section are missing.

This is an observant remark. Actually, we did not omit them by mistake. The reason for this being that we cannot measure emission quantum yields at low temperatures. This because our integrating sphere only functions at ambient conditions. Instead, we measured the intensity of emission at different temperatures. Then we note that the quantum yield of emission is proportional to the intensity of emission, and that the proportionality constant is temperature independent. By then comparing the relative rates as a function of temperature (i.e. $k(T)/k@77K$), this proportionality constant is canceled in the equation. By so doing, we can get information on the temperature dependence of k_{RISC} , and thus the activation energy for the process. The drawback is that we cannot compare the absolute difference in rates between samples. The methods section is updated as to clarify:

“It should be noted that the reason for using the proportionality between the intensity of emission and the emission quantum yields, is that emission from polaritonic states are dispersive and must therefore be measured in an integrating sphere. Ours is not compatible with cryogenic temperatures, with the solution to this dilemma being that relative rates is displayed in Figure 3a. This unable comparison of rates between samples, but captures the temperature dependence, and thus enables the calculation of energy barriers.”

Reviewer #3:

The paper reports an interesting application of cavity-controlled photo-physics, namely, the ability to reverse the usual energetic ordering of triplet and singlet states for organic molecules inserted into a tuned microcavity, thereby leading to barrierless reverse intersystem crossing, RISC. The authors present some rather convincing evidence of the effect for an organic chromophore DABNA-2: When the Rabi frequency is sufficiently large the lower polariton becomes positioned below the triplet state leading to an intense, delayed emission which monotonically decreases with temperature, reflecting the lack of an energy barrier that normally exists in RISC from neat films. However, if the lower polariton is detuned too far below the triplet, the barrierless RISC effectively shuts off, an effect the authors attribute to the lack of a substantial material component in the lower polariton. The paper is well-written, logically developed with solid, supportive evidence to back their hypothesis. I recommend publication after the following minor points are addressed.

1) I could not find in the text where the lifetime of the prompt emission in DABNA-2 neat films is reported. This is important because the delayed emission was always measured after waiting 100 nsec. I assume that 100 nsec is much greater than the fluorescence lifetime? The authors should make a point of this to ensure that the delayed fluorescence is entirely derived from RISC.

We would like to thank the reviewer for pointing out the need of conducting this important reference measurement. It is important to exclude the prompt emission part for the delayed emission study. The figure below shows time resolved prompt PL of a DABNA-2 neat film at 77 K (This experiment is conducted at low temperatures as to maximize the prompt emission lifetime). The prompt emission has decayed completely after about 50 ns. So the 100 nsec delay used in our experiments is enough to exclude any prompt contribution to the delayed emission. This new control measurement is incorporated into Supplementary Figure 2 (part f).

Supplementary Figure 2. Also seen is the prompt emission decay of a neat DABNA-2 film recorded at 77 K (f).

2) It would be very helpful if the phosphorescence peak in figure 2 was labelled and perhaps marked with a vertical line. The lower polariton emission peak should also be labelled.

The phosphorescence and polariton peaks has been labeled in accordance to the suggestion by the reviewer.

3) The authors need to better justify the cavity parameters. Importantly, why does cavity 4 have 50% larger Rabi frequency than cavity 1??? The Rabi frequency should scale as the sqrt of the molecular density. Is cavity 4 more dense?

We would like to thank the reviewer for this question, which forced us to go through our original data. The given Rabi splittings' in the original manuscript was wrong. The values given was the energy difference between the lower and upper polaritons at $k=0$. This error has been corrected, and the correct Rabi splittings' are about the same for all cavities (0.42, 0.42, 0.44 and 0.40 eV, for Cavity 1 to 4, respectively).

4) There should be some more discussion as to why the lower polariton in cavity 4 has a smaller material component than the polariton in cavity 3 despite cavity 4 having a larger Rabi frequency.

This questions stems from our erroneous values of the Rabi splittings' given in the original submission. Given the correct values in the updated manuscript, showing very similar Rabi splittings for all cavities, we believe that there is no need for such a discussion any more.

Reviewers' Comments:

Reviewer #1:

Remarks to the Author:

In the amended version of the manuscript, the authors have addressed the most part of reviewer's concerns. Especially, the new figure 5 is helpful in proving the validity of the main conclusion of this work: on the crossing of the triplet exciton state by a singlet exciton-polariton mode. As I mentioned in my first review, this effect has been sought for a long time. Potentially, it is important for applications of organic microcavities in opto-electronics and the realisation of organic lasers. For these reasons I recommend the manuscript for publication in its present form.

Reviewer #2:

Remarks to the Author:

The authors have resolved most of my original questions. With the phosphorescence clarified, I do think it is worth at least checking that these observations can't simply be explained by a Purcell enhancement in the phosphorescence rate when the cavity is approximately resonant with it near normal incidence in cavity3. Some of the observations are consistent with this possibility and, given the complexity of multiple competing rates (RISC, direct phosphorescence, T→LP scattering), all with different temperature dependences, it is not immediately clear what happens when the prefactor of one suddenly changes significantly in cavity3 (i.e. Purcell enhancement of the phosphorescence radiative rate) upsetting the balance between them, in conjunction with the claimed change in T→P- rate. In general, I would simply recommend that the authors go through a standard transfer matrix analysis given the bare film emission and confirm that their observations cannot be explained via the 'trivial' weak coupling effects from the cavity (i.e. change in radiative rate and redistribution of emission spectrum/angle).

To this last point, I would be fascinated to know what happens in the analog experiments for cavity1-4 carried out with a DABNA-2 concentration sufficiently decreased that they operate in the weak coupling regime (or with the same concentration and a thinner DABNA-2 layer, with extra space occupied by some transparent spacer layer). Does the cavity3 behavior no longer exhibit the barrierless DF emission rate once strong coupling is eliminated? I realize this is a large amount of additional work and would not hold up publication on this point, but it does make one curious to know.

Minor note: presumably the scale bar from Fig. 5a should be inverted given that it is reflectance. Fluorescence is also misspelled in the caption.

Reviewer #3:

Remarks to the Author:

I believe the authors addressed the reviewer's concerns appropriately. The manuscript should be published.

Reviewer #1 (Remarks to the Author):

In the amended version of the manuscript, the authors have addressed the most part of reviewer's concerns. Especially, the new figure 5 is helpful in proving the validity of the main conclusion of this work: on the crossing of the triplet exciton state by a singlet exciton-polariton mode. As I mentioned in my first review, this effect has been sought for a long time. Potentially, it is important for applications of organic microcavities in opto-electronics and the realisation of organic lasers. For these reasons I recommend the manuscript for publication in its present form.

Reviewer #2 (Remarks to the Author):

The authors have resolved most of my original questions. With the phosphorescence clarified, I do think it is worth at least checking that these observations can't simply be explained by a Purcell enhancement in the phosphorescence rate when the cavity is approximately resonant with it near normal incidence in cavity3. Some of the observations are consistent with this possibility and, given the complexity of multiple competing rates (RISC, direct phosphorescence, T→LP scattering), all with different temperature dependences, it is not immediately clear what happens when the prefactor of one suddenly changes significantly in cavity3 (i.e. Purcell enhancement of the phosphorescence radiative rate) upsetting the balance between them, in conjunction with the claimed change in T→P- rate. In general, I would simply recommend that the authors go through a standard transfer matrix analysis given the bare film emission and confirm that their observations cannot be explained via the 'trivial' weak coupling effects from the cavity (i.e. change in radiative rate and redistribution of emission spectrum/angle).

We agree with the reviewer that it is important to exclude Purcell enhancement. Since our transfer matrix code currently only can model absorbance (and it would take a significant time to implement and verify the correctness of the inclusion of an emitting layer in the code), we attacked this question through experiments. The figure below show the decay of prompt emission from a bare film and from Cavity3. The cavity energy of Cavity3 overlaps well with the molecular fluorescence. Furthermore, the envelopes of fluorescence and phosphorescence overlaps for this molecule. As a consequence, Purcell enhancement should be of equal magnitude both when measuring fluorescence and phosphorescence. The fitted lifetimes are within experimental error the same for the bare film and cavity samples. Thus, Purcell enhancement is negligible for both fluorescence and phosphorescence in Cavity3. Furthermore, in the standard picture of the emission from exciton polaritons, the very similar lifetimes indicate that decay from the exciton reservoir to the lower polariton occurs through radiative pumping.

Figure Caption: Emission decays of a neat film and Cavity3. Both excited using a laser diode with a centre wavelength of 425 nm.

To this last point, I would be fascinated to know what happens in the analog experiments for cavity1-4 carried out with a DABNA-2 concentration sufficiently decreased that they operate in the weak coupling regime (or with the same concentration and a thinner DABNA-2 layer, with extra space occupied by some transparent spacer layer). Does the cavity3 behavior no longer exhibit the barrierless DF emission rate once strong coupling is eliminated? I realize this is a large amount of additional work and would not hold up publication on this point, but it does make one curious to know.

This is a very interesting point, we will consider doing this experiment in coming studies.

Minor note: presumably the scale bar from Fig. 5a should be inverted given that it is reflectance. Fluorescence is also misspelled in the caption.

We would like to thank the reviewer for spotting these typos, which are now corrected.

Reviewer #3 (Remarks to the Author):

I believe the authors addressed the reviewer's concerns appropriately. The manuscript should be published.

Reviewers' Comments:

Reviewer #2:

Remarks to the Author:

The authors have satisfactorily addressed all of my questions. I recommend publication of this work in its current form.